# Investigation of the Interaction of Benzo(a)Pyrene and Fluoranthene with Cucurbit[n]urils (n = 6–8): Experimental and Molecular Dynamic Study

**DOI:** 10.3390/molecules28031136

**Published:** 2023-01-23

**Authors:** Abdalla A. Elbashir, Amira A. Alfadil, FakhrEldin O. Suliman, Ahmed O. Alnajjar

**Affiliations:** 1Department of Chemistry, College of Science, King Faisal University, Al-Ahsa 31982, Saudi Arabia; 2Department of Chemistry, Faculty of Science, University of Khartoum, Khartoum 11114, Sudan; 3Department of Chemistry, College of Science, Sultan Qaboos University, P.O. Box 36, Al-Khoud 123, Oman; 4Department of Chemical Laboratories, College of Science, Sudan University of Science and Technology, Khartoum 11115, Sudan

**Keywords:** Benzo(a)Pyrene, Cucurbit[n]uril, fluoranthen, inclusion complexes, molecular dynamics

## Abstract

The inclusion complexes of cucurbit[n]uril, CB[n] (n = 6–8), with poly aromatic hydrocarbon (PAH) Benzo(a)Pyrene (BaP), and fluoranthene (FLT) were investigated carefully in aqueous media. Fluorescence and ^1^H NMR spectroscopy were used to characterize and investigate the inclusion complexes that were prepared in the aqueous media. The most predominant complexes of both guests with hosts were the 1:1 guest: host complexes. Stability constants of 2322 ± 547 M^−1^, 7281 ± 689 M^−1^, 3566 ± 473 M^−1^ were obtained for the complexes of BaP with CB[6], CB[7], and CB[8], respectively. On the other hand, stability constants of 5900.270 ± 326 M^−1^, 726.87 ± 78 M^−1^, 3327.059 ± 153 M^−1^ were obtained for the complexes of FLT with CB[6], CB[7], and CB[8], respectively. Molecular dynamic (MD) simulations were used to study the mode and mechanism of the inclusion process and to monitor the stability of these complexes in aqueous media at an atomistic level. Analysis of MD trajectories has shown that both BaP and FLT form stable inclusion complexes with CB[7] and CB[8] in aqueous media throughout the simulation time, subsequently corroborating the experimental results. Nevertheless, the small size of CB[6] prohibited the encapsulation of the two PAHs inside the cavity, but stable exclusion complex was observed between them. The main driving forces for the stability of these complexes are the hydrophobic forces, van der Waals interactions, electrostatic effect, the π····π and C–H···π interaction. These results suggest that BaP and FLT can form stable complexes with CB[n] (n = 6–8) in solution.

## 1. Introduction

Polycyclic aromatic hydrocarbons (PAHs) are a large group of over 200 different compounds only made of carbon and hydrogen atoms containing two or more fused aromatic rings present in a mixture [1,2]. PAHs occur naturally in fossil fuels and as a byproduct of combustion or subjection processes involved in incineration and power generation [3]. They are nonpolar, with a highly hydrophobic and lipophilic nature, and occur as colorless, white/pale yellow solids with high melting and boiling points, and low vapor pressure [4]. PAHs are noxious and persistent organic pollutants, although they have low water solubility and exist in the environment at very low concentrations. They can easily cross the lipid membranes, are strongly bioaccumulative, and may interfere in normal DNA functioning. Hence, they are ecologically hazardous and can cause severe health hazards to humans, mainly in the form of cancer [5,6,7]. Since PAHs represent an emerging class of pollutants, and due to their recognized adverse effects, monitoring their levels is important in evaluating the risk associated with human consumption of food and various environmental exposure [5].

Benzo(a)Pyrene (BaP) is among the most carcinogenic PAHs. It is a molecule with five fused, hexagonal rings [8] (Figure 1d). BaP exists as a vapor at high temperatures or as adsorbed particulate, and it persists for long periods in the atmosphere in the particulate phase. The best way to remove BaP from the atmosphere is by photochemical oxidation [9]. BaP has a genotoxic nature, it can accumulate in various organs [10], it has been listed as a priority pollutant [11]. BaP is pleotropic, and can affect many cell- and organ-based systems. Hence, there are probably many modes of carcinogenic actions operating to a different extent in vivo [12]. BaP is widely used as a marker for total carcinogenic PAHs [13]. Fluoranthene (FLT) is ubiquitous in the environment and has been detected in air, foodstuff, and in drinking waters [14]. FLT has high thermal and air stability, and excellent fluorescent emission, as a result of which it has attracted intense attention in the area of optoelectronics, especially for organic light-emitting diodes [15].

Cucurbiturils CB[n]s are a family of macrocyclic compounds self-assembled from a condensation reaction of formaldehyde and glycoluril catalyzed by acid. These pumpkin-shaped containers have a characteristic rigid hydrophobic cavity accessible through two polar portals rimmed with carbonyl groups [16]. The cavity can bind with hydrophobic guest moieties [17]. Ion–dipole and dipole–dipole interactions, shape and size complementarity, the hydrophobic effect, hydrogen bonding, π-π interaction, and van der Waals interactions are the main driving factors that lead to the binding of guests with CB[n] [18]. Cucurbiturils cannot conform to the shape of guests, and therefore it is expected that complexation occurs with exceptional specificity and with high association constants [19]. This unique structure makes CB[n] a prime component in a wide range of applications, such as sensing, catalysis, biomolecular recognition, drug delivery, polymer material, molecular machines [20,21], separation of organic molecules [22] and in supramolecular assemblies [23].

Recently, considerable literature has grown up around using CB[n] as a host molecule; however, no single study exists on its complexation with an unsubstituted BaP and FLT. PAH detection procedures suffer from several major drawbacks; therefore, to validate a new more sensitive, selective, cheap, and easy analytical procedure, this paper shines new light on the influence of this macrocyclic molecule in the spectroscopic behavior of BaP and FLT. The complexes between the guests and these cavitands are developed in an attempt to enhance the electronic properties of the guests such as the fluorescence. This will allow the establishment of a platform for the development of simple direct analytical methods for their quantification in complex matrixes such as food sample.

The complexes in the solution were characterized by ^1^H NMR spectroscopy, and fluorescence spectroscopy. Furthermore, we utilized MD simulations in aqueous media to have a better insight into the major interactions between the guest and host and to identify the main factors stabilizing the complexes formed.

## 2. Experimental Section

### 2.1. General

All chemicals used in this study were purchased from Sigma Aldrich (St. Louis, MO, USA) and were used without further purification. The host’s CB[n] were synthesized based on literature methods [24], and their structures were verified using various spectroscopic techniques. Solvents used to prepare solutions are of HPLC grades and ultrapure water was used for aqueous solution preparation.

### 2.2. Fluorescence Measurement

The fluorescence spectra were obtained using a PerkinElmer LS55 fluorescence spectrophotometer (Perkin Elmer, Waltham, MA, USA) equipped with a xenon lamp using a quartz cell of 1.0 cm path length. All measurements were recorded at room temperature. The stock solutions of BaP and FLT were prepared in methanol, ultrapure water was used to prepare the working solutions, and the concentration of this solution was kept at 1.0 μM, while the concentration of hosts was varied from 0.1 to 0.6 mM for CB[n].

### 2.3. ^1^H NMR Spectroscopy

The nuclear magnetic resonance (NMR) spectroscopy experiments were carried out using a Bruker Avance Ⅲ HD 700 MHz spectrometer (Bruker, Karlsruhe, Germany) equipped with a 5 mm TCI H/C/N cryoprobe. The proton NMR experiment was run using the zg30 pulse program operating at 700.13 MHz. The acquisition parameters were as follows: 90° proton pulse width of 8.00 µs, relaxation delay of 1 s, and 128 scans. The spectrum was recorded at 298 K and processed using TOPSPIN 3.2 software (Bruker, Billerica, MA, USA). All host–guest complexes were prepared with DMSO-d6. All chemical shifts (expressed in ppm scale) are referenced to the solvent signal (DMSO-d6). ^1^H NMR data were processed with MestReNova v6.0.2-5475 software (Mestrelab Research S.L., Santiago de Compostela, Spain).

### 2.4. Molecular Modeling

The initial structure of BaP and FLT were built using Chemdraw and their geometries optimized using the DFT-B3LYP method using a 6-31G* basis set [25]. On the other hand, the structures of the hosts were extracted from the crystallographic parameters provided by the Structural Data Base System of the Cambridge crystallographic data center and were optimized by minimizing their energy using the PM6 semi-empirical method using MOPAC 2012 [26,27].

Molecular docking experiments were performed using the Autodock Iprogram (version 4.2) [28]. In this work, we used a Lamarckian genetic algorithm (LGA) for docking the guest into the host’s cavity to generate the inclusion complexes of the most appropriate conformation. Generally, Autodock defines the conformational space by implementing grids over all the possible search spaces. The initial torsions and positions of the guests were generated randomly, and Autodock tools were utilized to extract the optimum conformers using cluster analysis for all inclusion complexes by using a cutoff of 1.0 Å root mean square deviation (RMSD). The lowest-energy structures were further optimized using the PM6 semi-empirical model.

The MD simulations were carried out using the Desmond molecular simulations package, as distributed in the Schrodinger 2015 suite of programs. The OPLS_2005 all-atom force field with explicit solvent (TIP3P water model) was used throughout the calculations. Simulations were run with periodic boundary conditions, in an orthorhombic box with the solute, viz., placed in the middle at 20 Å distance from each of the box’s edges. The SHAKE algorithm was used to constrain covalent bonds between hydrogen and heavy atoms. Long-range electrostatic interactions were dealt with using the Ewald smooth particle mesh (PME) method [29]. The solvated molecules were subjected to sequential restraint solvent-solute minimizations and short MD simulations on NVT-NPT ensembles (as implemented in the default relaxation protocol in Desmond) coupled to the Berendsen thermostat. Finally, the production run was NPT run at 300 K and 1 bar. The simulations were then analyzed using the “simulation event analysis” module in the Schrodinger 2015 suite.

## 3. Result and Discussion

### 3.1. Fluorescence Measurement

The complex formation between BaP, FLT, and CB[n] (n = 6–8) hosts was studied carefully by the spectrofluorimetric technique. The fluorescence emission spectrum of BaP shows two intense characteristic vibronic structure emission bands at 405 and 420 nm and a broader less intense one at 540 nm when excited at 385 nm (Figure 2). At high BaP concentration, a structureless broad peak appears at 540 nm, we have noticed that this broad peak is vanishing at low concentration (less than 5 × 10^−7^ M) and upon complexation. According to the literature, some organic molecules tend to form aggregates in aqueous media through π-π type interactions between the aromatic rings [30]; specifically, BaP forms ground-state pairs that exhibit a redshift in the fluorescence spectrum compared to the concerning monomer [2]. Hence, we infer that this peak is for an excimer species.

The fluorescence emission spectrum of the polyaromatic compound BaP solution (5 × 10^−7^ M) displays a distinct enhancement on the 405 and 420 nm bands upon the addition of the three hosts, whereas the peak at 540 shows a monotonous increase in intensity with the increase in the hosts’ concentration (Figure 3). These results indicate that there is an interaction between BaP and the hosts. Previous research found that the formation of stable inclusion complexes is an exquisite or refined equilibrium between hydrophobicity, charge, shape, and size of the guest and host [31], and it is known that spectra of a fluorescent molecule may be affected when it is included or inserted inside a hydrophobic cavity [32]. Furthermore, the rigidity of the local environment and the polarity of the solvent affects the fluorescence emission spectra, while the nonpolar cavity also stabilizes the nonpolar excited state, resulting in emission at a lower energy or longer wavelength [33]. Consequently, in the presence of CB[n], the nonpolar B[a]P and FLT move from the polar aqueous media to the nonpolar environment (rigid cavity, or externally associate with the host).

The hydrophobic cavity of a cucurbituril encapsulates water molecules, which have high energy in accordance with their nonpolar cavities, and due to the weak dispersion interactions with weakly polarizable CB[n] cavity walls. Hence, the environment inside the cavity is not comfortable for these water molecules, which have stabilized themselves by hydrogen bonding interaction outside the cavity (bulk and portals). In return for that, the addition of hydrophobic molecules or moiety to the cavity and expulsion of high-energy water molecules is a highly exothermic process [34]. Moreover, the inclusion process results in the protection of the guest from the quenching effects of the bulk solvent and hence minimization of non-radiative decay pathways [35]. Among the three hosts, the enhancement of the fluorescence of BaP in the presence of CB[7] is higher than in the other hosts, even at a much lower concentration (Figure 3b). The size of CB[7] seems to be the most compatible with the guest size compared to the other hosts, hence comfortably encapsulating most of the BaP molecule. On the other hand, the polarity of the CB[7] solvent causes the BaP to be shunned from the medium to the hydrophobic host cavity, leading to high fluorescence intensity. CB[8] cavity volume is large enough (367 Å^3^) to encapsulate the BaP molecule comfortably and form a stable inclusion complex with high emission intensity; Figure 3c. The void space inside CB[8] may still keep some water molecules after the complexation, leading to a less nonpolar environment. Although CB[6] has a small cavity volume, the emission intensity of BaP and FLT increases strongly. CB[6] is typically unable to accommodate a six-membered aromatic ring [36]. Both PAHs are expected to form external supramolecular interactions with the macrocyclic CB[6], with high fluorescence enhancement [37]. In this case, the literature has interpreted these spectral properties in terms of the average environment surrounding the fluorophore [38], the existence of the guests in this new media reduces the degree of freedom of motion of them resulting in fluorescence enhancement, or the host may show surfactant properties [39]. This could lead to a change in the polarity of the guests’ solution or media.

In aqueous media, FLT shows excitation maxima at a wavelength of 350 nm, and a structureless broad emission peak that extends from 400 to 500 nm, centered on 455 nm (Figure 4). Significant enhancement of this broad emission band can be observed upon incremental additions of CB[n] aqueous solutions to 1.0 × 10^−6^ M FLT solution, as shown in Figure 5. This denotes an increase in the rigidity of the guest and a decrease in the energy wasted by internal conversion and collision with the solvent due to complex formation [31]. The increase in intensity became almost constant with increasing CB[6] at concentrations higher than 3.5 × 10^−4^ M, indicating an equilibrium state **(**Figure 5a) [40]. The large size of FLT compared to the CB[6] cavity volume possibly gives rise to an external association complex. The behavior of the FLT-CB[7] and FLT-CB[8] complexes are fairly similar, with similar emission enhancement trends; Figure 5b,c.

The concentration of the non-fluorescent host controls the enhancement of the guest’s fluorescence, according to the modified Bensi–Hilderbrand equation. Whereas $F_{\infty }$ and $F_{o}$ represent the fluorescence intensity in the maximum fluorescence intensity of the complex formed and the absence of host, respectively, F is the fluorescence intensity at each tested [host]_0_. K_a_ is an association or binding constant of the complex, it is a measurement of the thermodynamic stability of the host–guest complex at a given temperature in a given solvent [41]. The enhancement of the fluorescence intensity of the guests as a function of the added hosts fits well, assuming the formation of a 1:1 complex. The results are summarized in Table 1. A double reciprocal plot of the host concentration vs. 1/$\left(F_{\infty }-F_{o}\right)$ was produced to validate our assumption. (SI)
(1)$$\frac{1}{F-F_{o}}=\frac{1}{F_{\infty }-F_{o}}+\frac{1}{\left(F_{\infty }-F_{o}\right)\mathrm{Ka}{\left[M\right]}_{o}^{n}}$$

The variation in K_a_ values on all complexes reflects the effect of host size on the stability of the complexes. In addition to the host size, the packing coefficient (PC) can provide further explanation for this variation. PC is defined as the ratio of guest volume to the volume of the host cavity. It is considered one of the most important factors determining the stability of the host–guest complex. It works as an estimator of the steric goodness of fit of host–guest inclusion complexes. In general, stable inclusion complexes result when PC values are in the range from 45 to 65% [33,42]. Five-membered rings BaP perfectly match and pack the CB[7] cavity, giving a higher and more stable association constant than all of the other hosts. The large cavity of CB[8] allows the optimization of the water H-bond network to a degree that is structurally similar to bulk water, with the result that the energy contribution for water release is less from CB[8] than from CB[7] [34]. Despite its larger cavity, FLT superbly matches the CB[8] cavity. Surprisingly, the external CB[6]–guest complex is highly stable, too; the host molecules act as a solvent, and the dipole moment of the fluorophore interacts with the reactive fields induced in the surrounding solvent which led to the strength of the resulting complex [43].

### 3.2. ^1^H NMR Spectroscopy

NMR spectroscopy is a powerful tool to study the interactions of the host–guest molecules by providing detailed information on the molecular structure [44], and it can provide evidence for the inclusion of guests inside the host molecules by the monitoring the alteration of the chemical shift values before and after complexation and by the changes that might affect the shape of the proton signals. The proton’s chemical and the electronic environment of protons are affected by complexation with ensuing changes in the chemical shift values (δ) [39]. A significant change in the δ of the protons of both guest and host can be observed in the solution of the pure component as compared to the solution of the complex. The resonance signals of the BaP and FLT protons in the ^1^H NMR spectrum changed with the addition of the three hosts (Figure 6 and Figure 7). ^1^H NMR of pure BaP exhibit twelve different aromatic protons peaks ranging from 9.253 ppm–7.861 ppm, which are highly deshielded by the large anisotropic field generated by electron cloud in the ring’s π system [45]. The addition of CB[n] (n = 6–8) into the BaP solution induces spectral shifts suggesting changes in the surrounding electron or magnetic environment (Figure 6 and Figure 7). The induced changes in the proton chemical shifts for BaP in the presence of these macrocyclic hosts are summarized in Table 2 and Table 3. All BaP protons are shifted upfield upon complexation with CB[8], which means that the interior of the cavity provides magnetic shielding to the protons of the included guest. Nevertheless, CB[6] has a small cavity volume for B[a]P, but all B[a]P protons are shifted upfield in presence of this cavitand. BaP protons in its solution are highly deshielded by the π ring system of the neighboring BaP molecule (BaP exists in excimer form [30]). Disturbance of this excimer by addition of CB[6] shifted these protons to the lowest deshielding effect, hence, ∆δ < 0 was observed. With regard to CB[7], none of the protons underwent the same chemical shift (up- and downfield), this suggests that the chemical shift changes observed are due to ring current-induced shifts resulting from the reorientation of the rings A and D relative to the ring B, C, and E in the inclusion complex. One interesting finding is that all BaP peaks are sharp and well resolved in the presence of the hosts, reflecting a slower rate of exchange due to the inclusion complexation [46].

The ^1^H NMR of pure fluoranthene (FLT) exhibited five different aromatic peaks that were in good agreement with the documented values, ranging from 8.143 ppm to 7.438 ppm, which were shifted upon addition of the three hosts (Figure 7), giving evidence to support complexation [47]. It is worth mentioning here that FLT is a non-alternate aromatic hydrocarbon-containing angle strain that has different physicochemical behavior from an alternant one [48]. CB[8] is the only host that shifted the guest proton signal up-field, as shown in Figure 7d, hence the interior of the CB[8] cavity provides magnetic shielding to the protons of the included guest. The CB[8] cavity width is slightly too small to allow horizontal encapsulation of FLT, so it is diagonally located with respect to the axis of CB[8] [34]. Diederich et al. confirms that all protons in the direction perpendicular to the plane of the host are shifted up-field [49]. All FLT protons are shifted downfield after the addition of CB[6] and CB[7]; Figure 7b,c. This result is indicative of complex formation. The observed downfield shifts of the guest protons can be attributed to the anisotropic effect change in their surrounding environment [50]. After complexation, the radius of gyration of our guest is small which moves the guest protons more near the deshielding cone of the benzene ring and they will be affected by it more than the cavity hydrophobic shielding effect. The analysis of which specific guest protons are shifted up- or down-field provides important structural information about the nature and the specific mode of the inclusion. The question is: do parts or all the guests reside inside the cavity? And which side of the molecule is inserted into the cavity of the host? The value and sign of ∆δ are summarized in Table 4 and Table 5 could provide answers to these questions. All protons of the guest have been shifted with the same sign which is decided by the host used, for instance ∆δ is >0 for the FLT-CB[6]/CB[7] complexes and <0 for the FLT-CB[8] complex. This indicates that they are exposed to the same impact. As mentioned in the literature, the variations in hydrogen chemical shifts are mainly due to the difference in the C-C (π) and the C-C (σ) contributions [51].

### 3.3. Molecular Dynamic Simulation

Molecular dynamic (MD) simulation has not been applied for studying the interaction between BaP and FLT with CB[n]. Therefore, it was used here to predict the possibility of formation of BaP and FLT inclusion complexes with these hosts and to investigate their stability. MD provides valuable information pertinent to the fluctuations and conformational changes in atoms and molecules in materials [52]. The conformer with the lowest energy was obtained by docking the guest into the host’s cavity using the autodock program; then, after geometry optimization using the semispirical calculations, it was used as an input file for MD simulation. The docked structures were located in a box of TIP3P water, and were simulated at 300 K and 1 atm for 30 ns. The simulation event analysis module of the Schrodinger software was used to analyze the resultant trajectories and to calculate the RMSD and the radius of gyration (r_gyr_).

The r_gyr_ can be used to characterize the typical distance traveled by the body while the RMSD is used as a quantitative measure of the similarity between two or more complexes or molecular structures. BaP and FLT with CB[6] produce unstable trajectories, as is evident from the RMSD plots in Figure 8a and Figure 9a. The snapshots that were taken during the simulation time reveal that the guest molecule does not encapsulate inside the CB[6] cavity or even stay at the rim. It has been reported that from the electrostatic potential surface calculations on CBs, the outer surface is somewhat electrostatically positive. This could lead to interaction with the aromatic compounds via π∙∙∙π and/or C–H∙∙∙π interactions which are defined as outer-surface interactions [53].

The π∙∙∙∙π interaction is between the carbonyl group of CB[n] molecule and the PAH molecules, whereas C–H∙∙∙∙π interactions are between the methylene group on the outer surface of CB[n] molecules with the PAH molecules [16]. The RMSD results of the BaP-CB[8] complex shows less fluctuations compared to that of the BaP-CB[7] as shown in Figure 8b,c. Both complexes reach stability in less than 0.5 ns, and their RMSD values remain constant until the end of the simulation time. There are only minor differences in interactions within these two complexes, which stem mainly from the host cavity size. In both cases, the guest is well inserted into the host cavity, but the encapsulation of the BaP into CB[8] results in a structure more compact compared to BaP insertion into CB[7] as evident by the values of the rgyr. The snapshots illustrated in Figure 10 and Figure 11 for BaP-CB[7] and BaP-CB[8] complexes, respectively, show that BaP inserts the pyrene side into the cavity of these hosts.

FLT enters the cavity of CB[7] by pushing the fused two benzene rings aside (on the naphthalene side), as shown in Figure 12. An interesting behavior was encountered for the inclusion of FLT into the CB[8] cavity, as evidenced by the RMSD plot in Figure 9, with a huge increase in the RMSD value between 2.5 and 4 ns of the simulation time. The snapshot collected at 3 ns (Figure 13) shows that the complex remains intact with the guest molecule encapsulated inside the cavity of the host. After 4 ns, the inclusion complex of FLT-CB[8] shows a stable trajectory throughout the simulation. It is clear that most of FLT is inside the cavity to maximize the interactions with the host. The large cavity size of CB[8] allows this bulky guest to comfortably fit in it attaining the maximum stability as reflected in the higher binding energy.

The radius of gyration (r_gyr_) of the hosts, guests, and their corresponding inclusion complexes during the simulation time are summarized in Table 6 and Table 7. The values of the r_gyr_ of all complexes are comparable to those of the guests and are generally less than the sum of r_gyr_ of the individual host and guest. This indicates the existence of a compact association between the guest and the host, except in the case of CB[6].

## 4. Conclusions

Stable inclusion complexes of Benzo(a)Pyrene (BaP) and Fluranthene (FLT) were obtained with CB[n] (n = 6–8). The existence of the complexes was supported by fluorescence spectrophotometry, and ^1^H NMR spectroscopy. All results indicated the formation of 1:1 host: guest complexes. Packing coefficient PC and size complementarity plays a vital role in the complexation mechanism. Changes in the ^1^H NMR signals and δ value of the guest protons signify that the chemical and electronic environment of these protons is affected due to complexation. Molecular Dynamic MD simulations indicated that all complexes were stable in aqueous media during the simulation time. CB[6] forms an external association stable complex, but CB[7] and CB[8] form encapsulate the guests inside their cavity. Formation of diffraction patterns different from those recorded for pure hosts and guests are guaranteed evidence of inclusion complex formation.

## Figures and Tables

**Figure 1 molecules-28-01136-f001:**
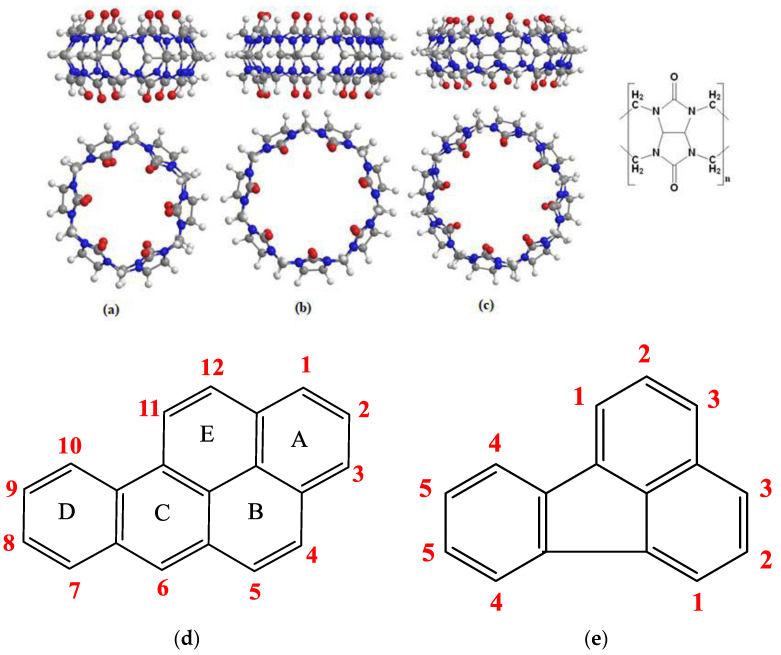
Top and side view of (**a**) CB[6], (**b**) CB[7], and (**c**) CB[8], general structure of CB[n] to the left. The structures of (**d**) Benzo(a) Pyrene and (**e**) Fluoranthene.

**Figure 2 molecules-28-01136-f002:**
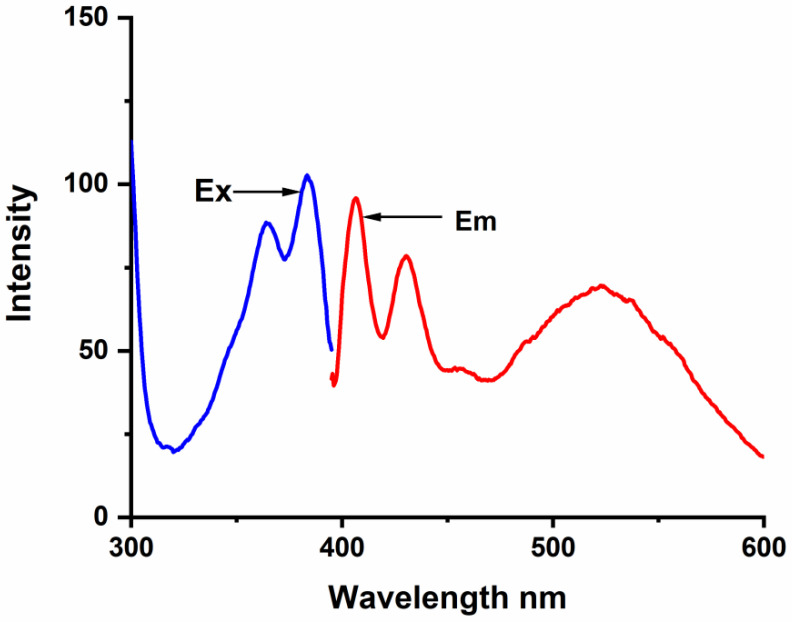
Excitation and emission spectra of Benzo(a)Pyren 1.0 × 10^−6^ M.

**Figure 3 molecules-28-01136-f003:**
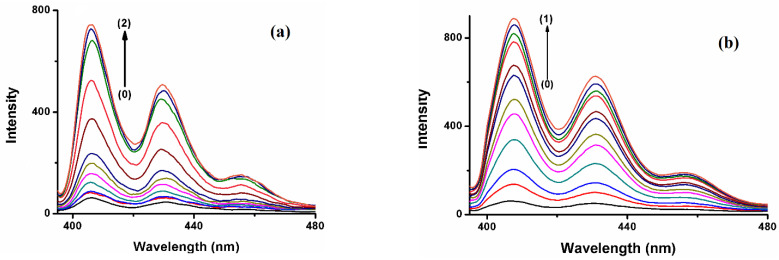
Fluorescence spectra of (0) B[a]P, 5.0 × 10^−7^ M with increasing concentration of (**a**) CB[6] 1 × 10^−5^ M → 2 × 10^−4^ M. (**b**) CB[7] 5.0 × 10^−6^ M → 0.8 × 10^−4^ M. (**c**) 0.2 × 10^−4^ M → 1.8 × 10^−4^ M.

**Figure 4 molecules-28-01136-f004:**
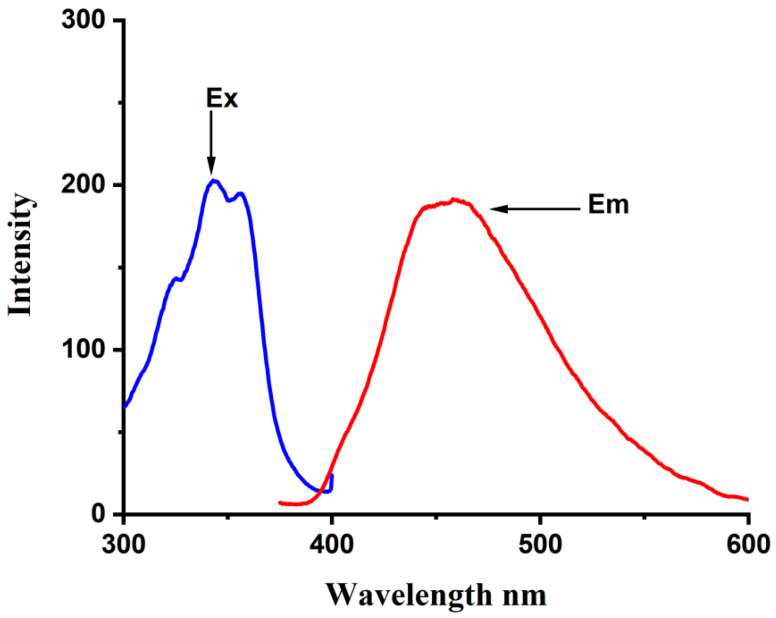
Excitation and emission spectra of Fluoranthene 1.0 × 10^−6^ M.

**Figure 5 molecules-28-01136-f005:**
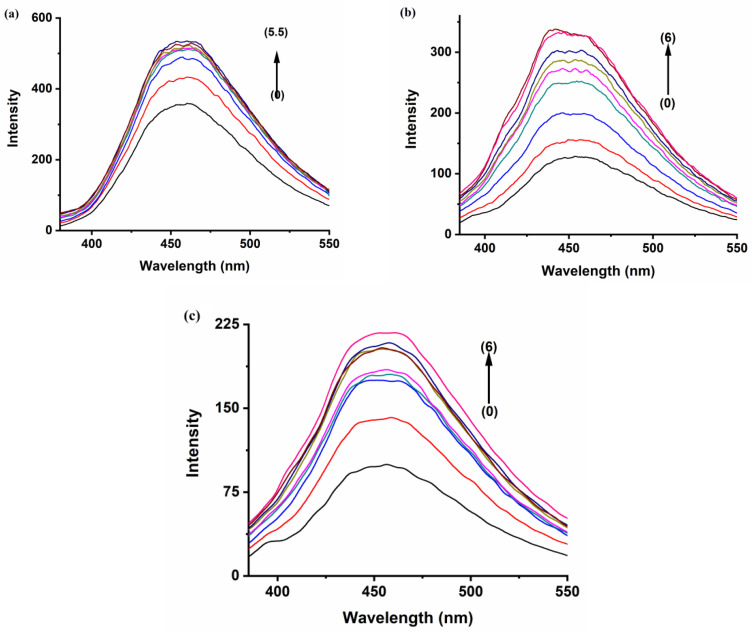
Fluorescence spectra of Fluoranthene (curve 0) 1.0 × 10^−6^ M with increasing concentrations of (**a**) CB[6] 1.0 × 10^−4^ M→5.5 × 10^−4^ M. (**b**) CB[7] 1.0 × 10^−4^ M→6.0 × 10^−4^ M. (**c**) CB[8] 1.0 × 10^−4^ M→6.0 × 10^−4^ M.

**Figure 6 molecules-28-01136-f006:**
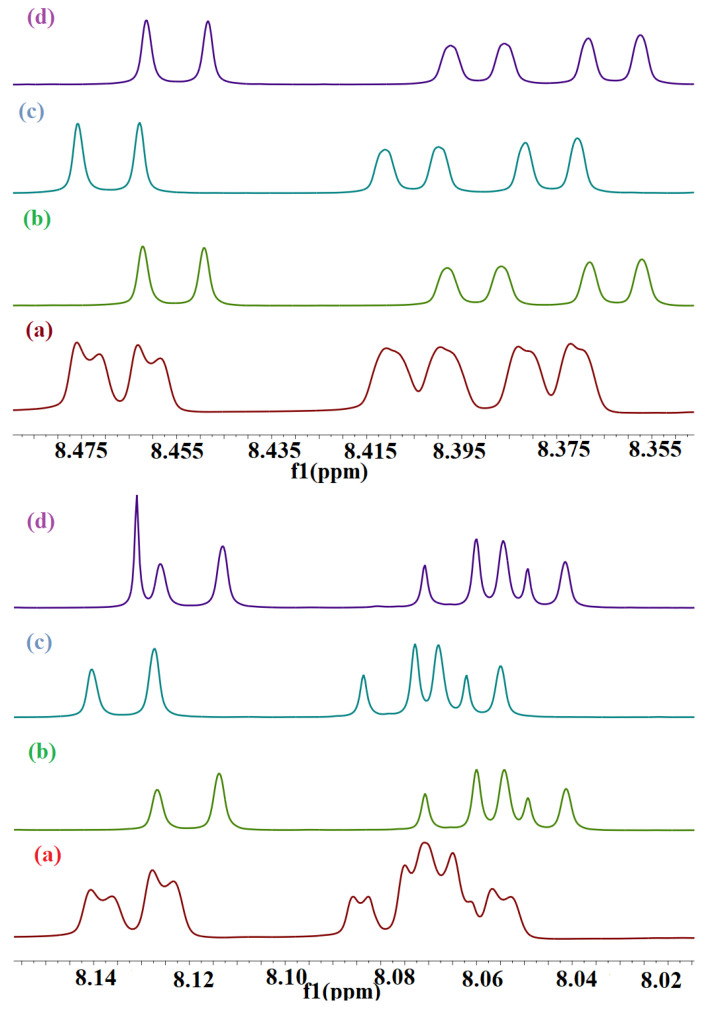
A partial ^1^H-NMR (700 MHz, DMSO-d6) spectra of BaP in the absence (**a**) and in the presence of (**b**) CB[6], (**c**) CB[7], and (**d**) CB[8]. The molar ratio of CB[n]:guest is 1:1.

**Figure 7 molecules-28-01136-f007:**
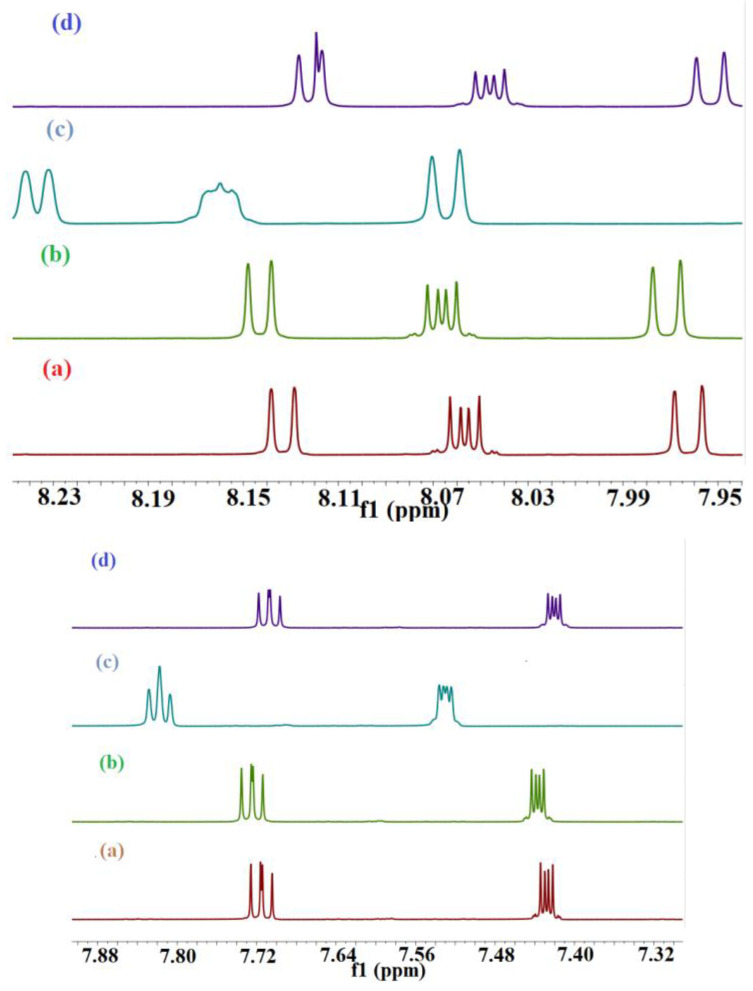
A partial ^1^H-NMR (700 MHz, DMSO-d6) spectra of FLT in the absence (**a**) and in the presence of (**b**) CB[6], (**c**) CB[7], and (**d**) CB[8]. The molar ratio of CB[n]:guest is 1:1.

**Figure 8 molecules-28-01136-f008:**
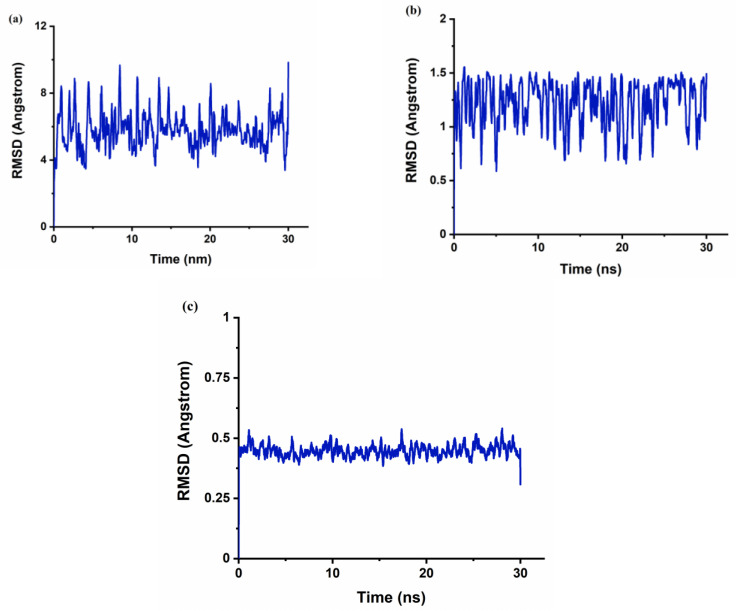
The time dependence of the root-mean-square deviations (RMSD) of atomic positions in the MD-simulated complexes of BaP with CB[n] from those in the corresponding energy minimized structure. (**a**) BaP-CB[6]; (**b**) BaP-CB[7]; (**c**) BaP-CB[8].

**Figure 9 molecules-28-01136-f009:**
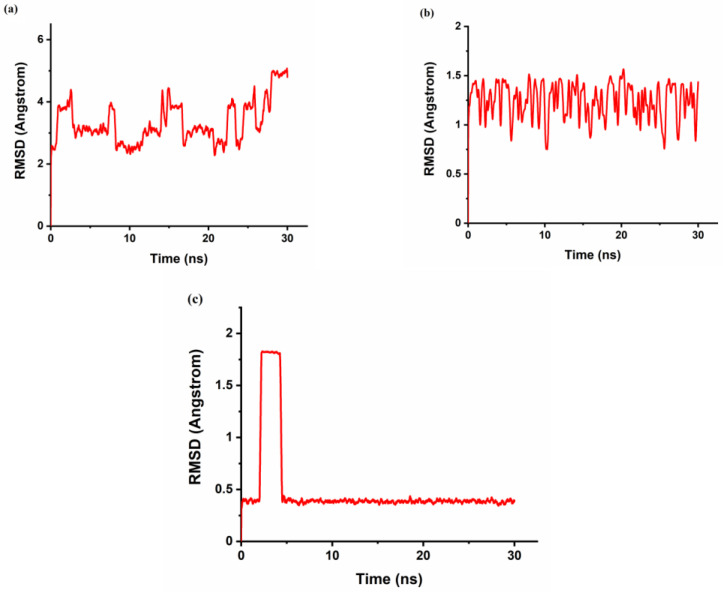
The time dependence of the root-mean-square deviations (RMSD) of atomic positions in the MD-simulated complexes of FLT with CB[n] from those in the corresponding energy minimized structure. (**a**) FLT-CB[6]; (**b**) FLT-CB[7]; (**c**) FLT-CB[8].

**Figure 10 molecules-28-01136-f010:**
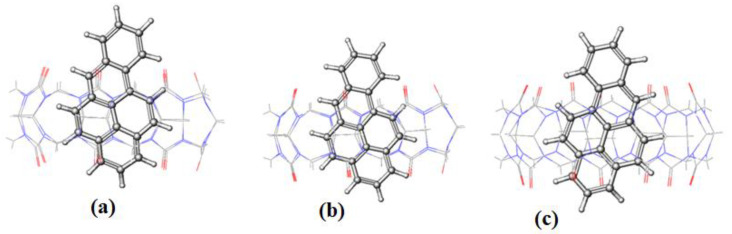
The representative snapshots of BaP-CB[7] obtained during simulation at (**a**) 0 ns; (**b**) 5 ns; (**c**) 30 ns.

**Figure 11 molecules-28-01136-f011:**
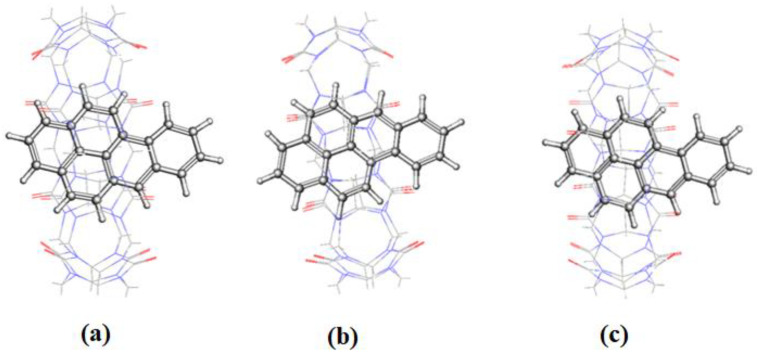
The representative snapshots of BaP-CB[8] obtained during simulation at (**a**) 0 ns; (**b**) 15 ns; (**c**) 30 ns.

**Figure 12 molecules-28-01136-f012:**
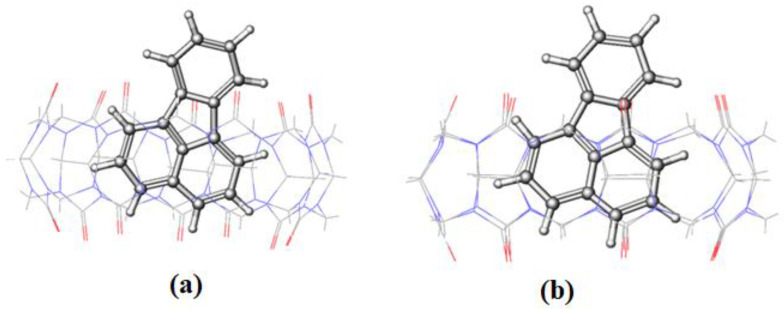
The representative snapshots of FLT-CB[7] obtained during simulation at (**a**) 0 ns and (**b**) 30 ns.

**Figure 13 molecules-28-01136-f013:**
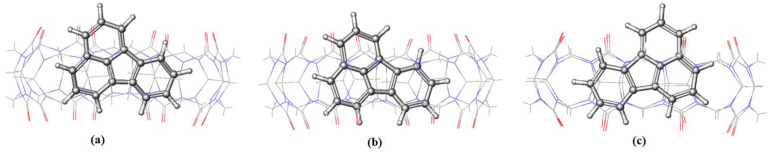
Representative snapshots of FLT-CB[8] obtained during simulation at (**a**) 0 ns; (**b**) 3 ns; (**c**) 30 ns.

**Table 1 molecules-28-01136-t001:** Stability constant of the complexation of CB[n] (n = 6–8) with BaP and FLT using Bensi–Hilderbrand equation.

Complexes	Binding Constant (M^−1^)	ΔG kJ mol^−1^
BaP-CB[6]	2322 ± 547	−19.00
BaP-CB[7]	7281 ± 689	−21.81
BaP-CB[8]	3566 ± 473	−20.06
FLT-CB[6]	5900 ± 326	−21.30
FLT-CB[7]	727 ± 78	−16.16
FLT-CB[8]	3327 ± 153	−19.89

**Table 2 molecules-28-01136-t002:** ^1^H-NMR shifts between free and complexed protons in BaP-CB[6]/CB[8] complex.

Proton	δ_(Free)_/ppm	δ_(BaP-CB[6])_/ppm	∆δ_(δComplex-δFree)_/ppm	δ_(BaP-CB[8])_/ppm	∆δ_(δComplex-δFree)_/ppm
H1′	8.404	8.392	−0.012	8.391	−0.013
H2′	8.228	8.214	−0.014	8.215	−0.013
H3′	8.376	8.363	−0.014	8.363	−0.013
H4′	8.070	8.054	−0.016	8.054	−0.016
H5′	8.132	8.120	−0.012	8.119	−0.013
H6′	8.717	8.706	−0.012	8.705	−0.012
H7′	8.460	8.449	−0.011	8.449	−0.012
H8′	7.861	7.848	−0.013	7.848	−0.013
H9′	7.903	7.892	−0.011	7.890	−0.013
H10′	9.253	9.233	−0.020	9.235	−0.018
H11′	9.228	9.208	−0.020	9.210	−0.018
H12′	8.473	8.462	−0.011	8.462	−0.011

**Table 3 molecules-28-01136-t003:** The ^1^H NMR shifts between free and complexed protons in BaP-CB[7] complex.

Proton	δ_(Free)_/ppm	δ_(BaP-CB[7])_/ppm	∆δ_(δComplex-δFree)_/ppm
H1′	8.404	8.405	−0.003
H2′	8.228	8.228	−0.003
H3′	8.376	8.376	0.003
H4′	8.070	8.078	0.004
H5′	8.132	8.134	0.004
H6′	8.717	8.720	0.001
H7′	8.460	8.463	0.000
H8′	7.861	7.860	−0.001
H9′	7.903	7.903	0.002
H10′	9.253	9.250	0.008
H11′	9.228	9.225	0.008
H12′	8.473	8.476	−0.001

**Table 4 molecules-28-01136-t004:** ^1^H NMR shifts between free and complexed protons in FLT-CB[6] and CB[7].

Proton	δ_(Free)_/ppm	δ_(FLT-CB[6])_/ppm	∆δ_(δComplex-δFree)_/ppm	δ_(FLT-CB[7])_/ppm	∆δ_(δComplex-δFree)_/ppm
H1′	8.133	8.143	0.010	8.235	0.103
H2′	7.715	7.775	0.060	7.818	0.103
H3′	7.962	7.972	0.010	8.064	0.102
H4′	8.057	8.066	0.009	8.160	0.103
H5′	7.428	7.438	0.010	7.530	0.103

**Table 5 molecules-28-01136-t005:** ^1^H NMR shifts between free and complexed protons in the FLT-CB[8] complex.

Proton	δ_(Free)_/ppm	δ_(FLT-CB[8])_/ppm	∆δ_(δComplex-δFree)_/ppm
H1′	8.133	8.118	−0.015
H2′	7.715	7.708	−0.008
H3′	7.962	7.953	−0.009
H4′	8.057	8.046	−0.011
H5′	7.428	7.420	−0.008

**Table 6 molecules-28-01136-t006:** Average value of RMSD and radius of gyration obtained from molecular dynamics trajectories for various species.

Compound	RMSD (Å)	r_gyr_ (Å)
BaP-CB[6]	5.8 ± 2.12	7.81 ± 3.34
BaP	0.2 ± 0.04	3.18 ± 0.01
CB[6]	0.3 ± 0.07	4.82 ± 0.02
BaP-CB[7]	1.2 ± 0.38	5.21 ± 0.02
BaP	0.2 ± 0.04	5.53 ± 0.02
CB[7]	1.0 ± 0.42	3.18 ± 0.01
BaP-CB[8]	0.5 ± 0.10	5.71 ± 0.02
BaP	0.21 ± 0.04	3.18 ± 0.01
CB[8]	0.43 ± 0.10	6.06 ± 0.02

**Table 7 molecules-28-01136-t007:** Average value of RMSD and radius of gyration obtained from molecular dynamics trajectories for various species.

Compound	RMSD (Å)	r_gyr_ (Å)
FLT-CB[6]	3.34 ± 0.73	5.51 ± 0.08
FLT	0.20 ± 0.03	2.74 ± 0.01
CB6	0.34 ± 0.07	4.82 ± 0.02
FLT-CB[7]	1.24 ± 0.36	5.23 ± 0.02
FLT	0.16 ± 0.03	2.73 ± 0.01
CB7	0.97 ± 0.37	5.52 ± 0.02
FLT-CB[8]	0.49 ± 0.38	5.76 ± 0.02
FLT	0.18 ± 0.03	2.73 ± 0.01
CB8	0.47 ± 0.09	6.08 ± 0.02

## Data Availability

Not applicable.

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
