# Peer review of "Investigation of the Interaction of Benzo(a)Pyrene and Fluoranthene with Cucurbit[n]urils (n = 6–8): Experimental and Molecular Dynamic Study"

_molecules, 2023, doi:10.3390/molecules28031136_

Round 1

Reviewer 1 Report

Comments on the manuscript molecules-2138763:

The manuscript with reference number: molecules-2138763 entitled " Investigation of the interaction of Benzo(a)pyrene and Fluoranthene with cucurbit[n]urils (n = 6-8): Experimental and Molecular Dynamic Study " studied the interactions between mazaquinpoly aromatic hydrocar- 13 bon (PAH) benzo(a)pyrene (BaP), and fluoranthene (FLT) and CB[n]s (n = 6-8). Moreover, the method of molecular dynamics simulation is also used to analyze the mode and mechanism of the inclusion process and to monitor the stability of these complexes. However, there are many due tasks that have not been completed. Thus, this manuscript can be published in journals after major revision.

(1) The authors characterized six new complexes based on CB[n]s and BaP and FLT. The novelty of the presented complex is beyond doubt. However, the purpose of the study remains unclear. The authors point to the environmental significance of PAHs. Does this mean that the resulting complexes can be used for wastewater treatment, drug delivery or other similar applications? The authors point to several major drawbacks of PAHs detection procedures and to validate a new more sensitive, selective, cheap, and easy analytical procedure for PAHs detection in the introduction. However, they did not mention this in any way in the subsequent parts of the article. The authors should comment on this.

(2) " The complexes in solution were characterized by 1H NMR spectroscopy, mass spectrometry, and fluorescence spectroscopy". However, the authors did not perform mass spectrometry in the article.

(3) " The inclusion complexes ... in aqueous media and in the solid state". The solid-state state of the complexes was not investigated in the article.

(4) " the main driving force ... is thermodynamically controlled through enthalpy-entropy compensation". The driving force cannot be seen from fluorescence spectroscopy.

(5) CB[n]s (n=6-8) are insoluble in DMSO, but 1H NMR is carried out in DMSO.

(6) The fluorescence spectroscopy experiments were performed in aqueous solution, but the 1H NMR experiments were performed in DMSO-d6.

(7) This reviewer found it difficult to follow the Figures on account of insufficient information included in the figures and or captions. For example, it is better to include the ratios of CB[n]s and BaP and FLT in Figures 6 and 7. In Figure 6, the illustrations of (a)-(d) are unclear.

(8) Molecular dynamics simulation results are inconsistent with the NMR results. Meanwhile, molar ratio of complexes should also be taken into account as well.

(9) There are multiple format issues. For example, In "1H NMR", the number "1" should be superscript. " CBn, CB(n]" should be "CB[n]". "BaP" or "B(a)P", please unify the abbreviation format. Please make sure that the font is the same in all the figures in the article. Please check the title ".2 Experimental" and " 2.3.1. H NMR spectroscopy".

(10) Some related CB[n] references are recommended to be cited: 1) Angewandte Chemie International Edition, 2022, 61, e 202207209; 2) Chinese Chemical Letters, 2022, 108040; 3) Chinese Chem. Lett., 2022, 33, 24552458; 4) Separation and Purification Technology, 2023, 304, 122342.

Author Response

Review Report (Reviewer 1)

The manuscript with reference number: molecules-2138763 entitled " Investigation of the interaction of Benzo(a)pyrene and Fluoranthene with cucurbit[n]urils (n = 6-8): Experimental and Molecular Dynamic Study " studied the interactions between mazaquinpoly aromatic hydrocar- 13 bon (PAH) benzo(a)pyrene (BaP), and fluoranthene (FLT) and CB[n]s (n = 6-8). Moreover, the method of molecular dynamics simulation is also used to analyze the mode and mechanism of the inclusion process and to monitor the stability of these complexes. However, there are many due tasks that have not been completed. Thus, this manuscript can be published in journals after major revision.

  • The authors characterized six new complexes based on CB[n]s and BaP and FLT. The novelty of the presented complex is beyond doubt. However, the purpose of the study remains unclear. The authors point to the environmental significance of PAHs. Does this mean that the resulting complexes can be used for wastewater treatment, drug delivery or other similar applications? The authors point to several major drawbacks of PAHs detection procedures and to validate a new more sensitive, selective, cheap, and easy analytical procedure for PAHs detection in the introduction. However, they did not mention this in any way in the subsequent parts of the article. The authors should comment on this.

We added this statement in the introduction: paragraph 4 lines 5- 8.

The complexes between the guests and these cavitands are developed in an attempt to enhance the electronic properties of the guests such as the fluorescence. This will allow the establishment of a platform for the development of simple direct analytical methods for their quantification in complex matrixes such as food sample.

  • " The complexes in solution were characterized by 1H NMR spectroscopy, mass spectrometry, and fluorescence spectroscopy". However, the authors did not perform mass spectrometry in the article. 

Have been corrected in the text and mass spectrometry was deleted.

  • " The inclusion complexes ... in aqueous media and in the solid state". The solid-state state of the complexes was not investigated in the article.

The solid state is deleted.

  • " the main driving force ... is thermodynamically controlled through enthalpy-entropy compensation". The driving force cannot be seen from fluorescence spectroscopy.

This statement has been removed from the text. However, from the literature similar association were found enthalpy driven e.g. (Greene et al., 2017) (Tang et al., 2019). Moreover, we included a column in Table 1 for the free energy change.

  • CB[n]s (n=6-8) are insoluble in DMSO, but 1H NMR is carried out in DMSO.

In the presence of the guests, they dissolve to some extent in DMSO. Our guests are not soluble in D2O, they are more soluble in DMSO. Also, there are studies showing 1HNMR of CB[n] in DMSOd6 (Investigation of inclusion complexes of ametryne and atrazine with cucurbit[n]urils (n=6–8) using experimental and theoretical techniques. (L. Araújo de Azevedo et al., Photochem. Photobiol. Sci.,2017,16, 663–671; M. S. Mokhtar F. O. Suliman· A. A. Elbashir, J. Incl. Phenom. Macrocycl. Chem., 94, pages31–43 (2019).

  • The fluorescence spectroscopy experiments were performed in aqueous solution, but the 1H NMR experiments were performed in DMSO-d6.

The fluorescence was studied in water media because the study is targeting applications in aqueous environment. DMSO-d6 gave better 1H NMR results compared to D2O.

  • This reviewer found it difficult to follow the Figures on account of insufficient information included in the figures and or captions. For example, it is better to include the ratios of CB[n]s and BaP and FLT in Figures 6 and 7. In Figure 6, the illustrations of (a)-(d) are unclear.

Have been corrected in the text on page 15 and 17.

  • Molecular dynamics simulation results are inconsistent with the NMR results. Meanwhile, molar ratio of complexes should also be taken into account as well.

The differences in the media may account for this difference. The ratio of the guest to the host is always kept 1:1

  • There are multiple format issues. For example, In "1H NMR", the number "1" should be superscript. " CBn, CB(n]" should be "CB[n]". "BaP" or "B(a)P", please unify the abbreviation format. Please make sure that the font is the same in all the figures in the article. Please check the title ".2 Experimental" and " 2.3.1. H NMR spectroscopy".

Have been corrected in the text.

(10) Some related CB[n] references are recommended to be cited: 1) Angewandte Chemie International Edition, 2022, 61, e 202207209; 2) Chinese Chemical Letters, 2022, 108040; 3) Chinese Chem. Lett., 2022, 33, 2455–2458; 4) Separation and Purification Technology, 2023, 304, 122342.

Have been cited in the text on page 5 reference number 22 and 23.

Reviewer 2 Report

1) Some of the references are in superscript change them to normal mode like references 1,5 and 30.

2) Kindly provide optimal concentration at which 520 nm disappears on complexation in section 3.1.

3) Also compute free energy changes in Table 1 from binding constants.

4) Why chemical shift change is negative for FLT-CB[8] complex please explain.

5) Please cite proper references for PM3 method:

Stewart, James J. P. (1989). "Optimization of parameters for semiempirical methods I. Method". J. Comput. Chem. 10 (2): 209–220.

ALSO JUSTIFY THEIR USE HERE.

6) Cite some references as following to ensure that B3LYP is worth using for organic molecules computations.

Wasif Baig, M., Pederzoli, M., Kyvala, M., Cwiklik, L. and Pittner, J., 2021. Theoretical Investigation of the Effect of Alkylation and Bromination on Intersystem Crossing in BODIPY-Based Photosensitizers. The Journal of Physical Chemistry B, 125(42), pp.11617-11627.

Isborn, C.M., Luehr, N., Ufimtsev, I.S. and Martínez, T.J., 2011. Excited-state electronic structure with configuration interaction singles and Tamm–Dancoff time-dependent density functional theory on graphical processing units. Journal of Chemical Theory and Computation, 7(6), pp.1814-1823.

7) Also cite recent literature for aggregation among molecules in introduction like:

Lunkad, R., Barroso da Silva, F.L. and Kosovan, P., 2022. Both Charge-Regulation and Charge-Patch Distribution Can Drive Adsorption on the Wrong Side of the Isoelectric Point. Journal of the American Chemical Society, 144(4), pp.1813-1825.

Author Response

Review Report (Reviewer 2)

1) Some of the references are in superscript change them to normal mode like references 1,5 and 30.

Have been corrected in the text.

2) Kindly provide optimal concentration at which 520 nm disappears on complexation in section 3.1.

Have been illustrated in the text on page 7, lines number 18 and 19.

3) Also compute free energy changes in Table 1 from binding constants.

Have been computed and added to Table 1, on page 13.

4) Why chemical shift change is negative for FLT-CB[8] complex please explain.

It was explained in page 15, line number 39 and page 16 line1.

5) Please cite proper references for PM3 method:

Stewart, James J. P. (1989). "Optimization of parameters for semiempirical methods I. Method". J. Comput. Chem10 (2): 209–220.

 Have been cited in the text on page 6 reference number 25.

6) Cite some references as following to ensure that B3LYP is worth using for organic molecules computations.

Wasif Baig, M., Pederzoli, M., Kyvala, M., Cwiklik, L. and Pittner, J., 2021. Theoretical Investigation of the Effect of Alkylation and Bromination on Intersystem Crossing in BODIPY-Based Photosensitizers. The Journal of Physical Chemistry B125(42), pp.11617-11627.

Isborn, C.M., Luehr, N., Ufimtsev, I.S. and Martínez, T.J., 2011. Excited-state electronic structure with configuration interaction singles and Tamm–Dancoff time-dependent density functional theory on graphical processing units. Journal of Chemical Theory and Computation7(6), pp.1814-1823.

 Have been cited in the text on page number 6 reference number 24.

7) Also cite recent literature for aggregation among molecules in introduction like:

Lunkad, R., Barroso da Silva, F.L. and Kosovan, P., 2022. Both Charge-Regulation and Charge-Patch Distribution Can Drive Adsorption on the Wrong Side of the Isoelectric Point. Journal of the American Chemical Society144(4), pp.1813-1825.

Kindly accept our apology for citing this reference.

Reviewer 3 Report

In this manuscript, the authors investigated the complexes of cucurbit[n]uril with benzo(a)pyrene and fluoranthene with the aim of developing a simple and inexpensive method for the detection of these compounds. The complexes were characterized by experimental measurements (fluorescence spectrophotometry and NMR) and computational modeling (MD simulations).

The complexes are adequately characterized and the conclusions are convincing. The overall impression of the manuscript is positive and the manuscript is certainly valuable for chemists working in this field. In my opinion, the manuscript can be published after a minor revision:
Something is wrong with the numbering of the tables and figures. Tables 5 and 6 should be 4 and 5. What do the figures on pages 10 and 12 represent? They are not labeled and are part of Figures 6 and 7. Please clarify.

Author Response

Review Report (Reviewer 3)

In this manuscript, the authors investigated the complexes of cucurbit[n]uril with benzo(a)pyrene and fluoranthene with the aim of developing a simple and inexpensive method for the detection of these compounds. The complexes were characterized by experimental measurements (fluorescence spectrophotometry and NMR) and computational modeling (MD simulations).

The complexes are adequately characterized and the conclusions are convincing. The overall impression of the manuscript is positive and the manuscript is certainly valuable for chemists working in this field. In my opinion, the manuscript can be published after a minor revision:
Something is wrong with the numbering of the tables and figures. Tables 5 and 6 should be 4 and 5. What do the figures on pages 10 and 12 represent? They are not labeled and are part of Figures 6 and 7. Please clarify.

All reviewer’s remarks have been corrected in the text. See pages 11, 15, 17,18.
